# Facilitation or Competition? Effects of Lions on Brown Hyaenas and Leopards

**Janelle Bashant [1], Michael Somers [2], Lourens Swanepoel [3] and Fredrik Dalerum [1,4,5,\*]**

[1] Mammal Research Institute, Department of Zoology and Entomology, University of Pretoria, Private Bag X20, Pretoria 0028, South Africa; nelthebell86@hotmail.com

[2] Centre for Invasion Biology, Eugène Marais Chair of Wildlife Management, Mammal Research Institute, Department of Zoology and Entomology, University of Pretoria, Private Bag X20, Pretoria 0028, South Africa; michael.somers@up.ac.za

[3] Department of Zoology, University of Venda, Private Bag X5050, Thohoyandou 0950, South Africa; lourens.swanepoel.univen@gmail.com

[4] Department of Zoology, Stockholm University, 10691 Stockholm, Sweden

[5] Research Unit of Biodiversity (UMIB, UO-CSIC-PA), Campus Mieres, Edificio de Investigación, 33600 Mieres, Spain

**\*** Correspondence: fredrik.dalerum@csic.es

**Abstract:** Intra-guild interactions related to facilitation and competition can be strong forces structuring ecological communities and have been suggested as particularly prominent for large carnivores. The African lion (*Panthera leo*) is generally thought to be a dominant predator where it occurs and can be expected to have broad effects on sympatric carnivore communities. We used data from two small game reserves in northern South Africa to relate the presence of African lions to abundance, habitat use, diet, and prey selection of two sympatric large carnivores, brown hyaenas (*Parahyaena brunnea*) and leopards (*Panthera pardus*). Our results offered some support for the facilitative effects of lions on brown hyaenas, and competitive effects on leopards. However, differences between populations living without and with lions were restricted to broad diet composition and appear not to have permeated into differences in either prey selection, abundance or habitat use. Therefore, we suggest that the potential effects of lions on the predator–prey interactions of sympatric predators may have been context dependent or absent, and subsequently argue that lions may not necessarily influence the predator–prey dynamics in the landscapes they live in beyond those caused by their own predatory behaviour.

**Keywords:** community ecology; predation; interference competition; landscape of fear; apex predator; Carnivora; African lion; African leopard; brown hyaena

## 1. Introduction

To fully understand the dynamics of ecological communities, it is critically important to quantify interactions among individual species [1]. The guild concept has been suggested as a useful way to categorise species into classes of organisms that share the same resources, typically without regard to taxonomic affiliation, to better understand processes related to resource partitioning between species [2]. The combined interspecific interactions within guilds have been suggested to strongly influence the structuring of ecological communities [3]. Species present within a specific guild can be subjected to both competition and facilitation by other members of that guild. Competition arises when two species share similar resources in areas where they co-exist [4]. However, the nature of interactions between species in a guild is not necessarily competitive but can also be facilitative and positively benefit one or both species [5]. Intra-guild competition can be a powerful evolutionary force and change a species'

resource use to lessen the negative effects inflicted by competitive guild members [6,7]. Contrarily, facilitation may have positive effects between sympatric species by increasing the possibility to utilise certain resources [8].

Intra-guild interactions have the potential to be particularly prominent among large mammalian carnivores, since they generally have large resource overlaps [9,10]. Interspecific competition may also be more severe in carnivore guilds compared to other vertebrate groups due to their morphological adaptations to kill prey [11–13]. Large carnivore guilds are typically dominated by a single keystone species, which has a larger ecosystem impact compared to the other species in the guild [14,15]. However, facilitative interactions are also common, primarily through the provisioning of carcasses [8]. Larger dominant carnivores can hunt and kill large prey leading to larger sizes of carcasses left for opportunistic guild mates. Due to the availability of a higher diversity of food for species that mostly scavenge in communities where dominant carnivore species occur, an increase in their dietary breadth would be expected.

Africa contains one of the few remaining intact large predator guilds in the world [16], where the lion (*Panthera leo*) is the dominant species where it occurs [15]. The brown hyaena (*Parahyaena brunnea*) and the leopard (*Panthera pardus*) are two less dominant African carnivores that may experience different consequences of living together with lions. The brown hyaena has generally been described as an opportunistic forager, although it mostly has been shown to feed on ungulates [17,18]. The leopard is, in contrast, a specialist predator on small- and medium-sized prey [19]. Since lions are specialised predators on medium-sized and large vertebrates [20], we can expect facilitative interactions between lions and brown hyaenas due to carcass provisioning, and competitive interactions between lions and leopards due to both species' predatory habits. However, recent studies suggest that leopards may be unaffected by lions in terms of their abundance, space use, activity and prey utilisation [21–23], although some observations suggest spatial avoidance of lions by leopards [24]. Relationships between lions and brown hyaenas are less well documented, but facilitative interactions have been suggested [18,25].

In this study, we contrasted the abundance, habitat use, diet and prey selection of brown hyaenas and leopards between areas without and with lions present to explore the nature of interspecific interactions between lions and these two large African carnivores. We predicted that brown hyaenas, due to facilitative interactions primarily related to carrion provisioning, would show increased abundance in the presence of lions, but that low levels of intraspecific competition would result in limited differences in habitat use for populations living with and without lions. We also predicted that brown hyaenas living in the presence of lions would feed more on large ungulates, presumably from carrion. For leopards, on the other hand, we predicted that competition with lions would result in lower leopard abundance in the presence of lions that leopards would shift their habitat use in response to lion presence, and that leopards would feed on smaller-sized prey in the presence of lions.

## 2. Materials and Methods

### 2.1. Study Area

The study was conducted on two nature reserves in the Waterberg region of South Africa (Figure 1A), Lapalala Wilderness (Lapalala, 23°54′00′′ S, 28°19′23′′ E) and Welgevonden Private Game Reserve (Welgevonden, 24°19′23′′ S, 28°02′37′′ E). The Waterberg falls into the summer rainfall region with a mean annual rainfall of 500 mm and is climatically defined by a dry winter season from May to October and a wet summer season from November to April [26,27]. Temperatures range from 14°C to 30 °C during the summer and from 2°C to 30 °C during winter, with frosts often present at night. The area is characterised by nutrient-poor soils with rich patches, cliffs and deep valleys [28], and the vegetation is classed as Waterberg Mountain Bushveld [29].

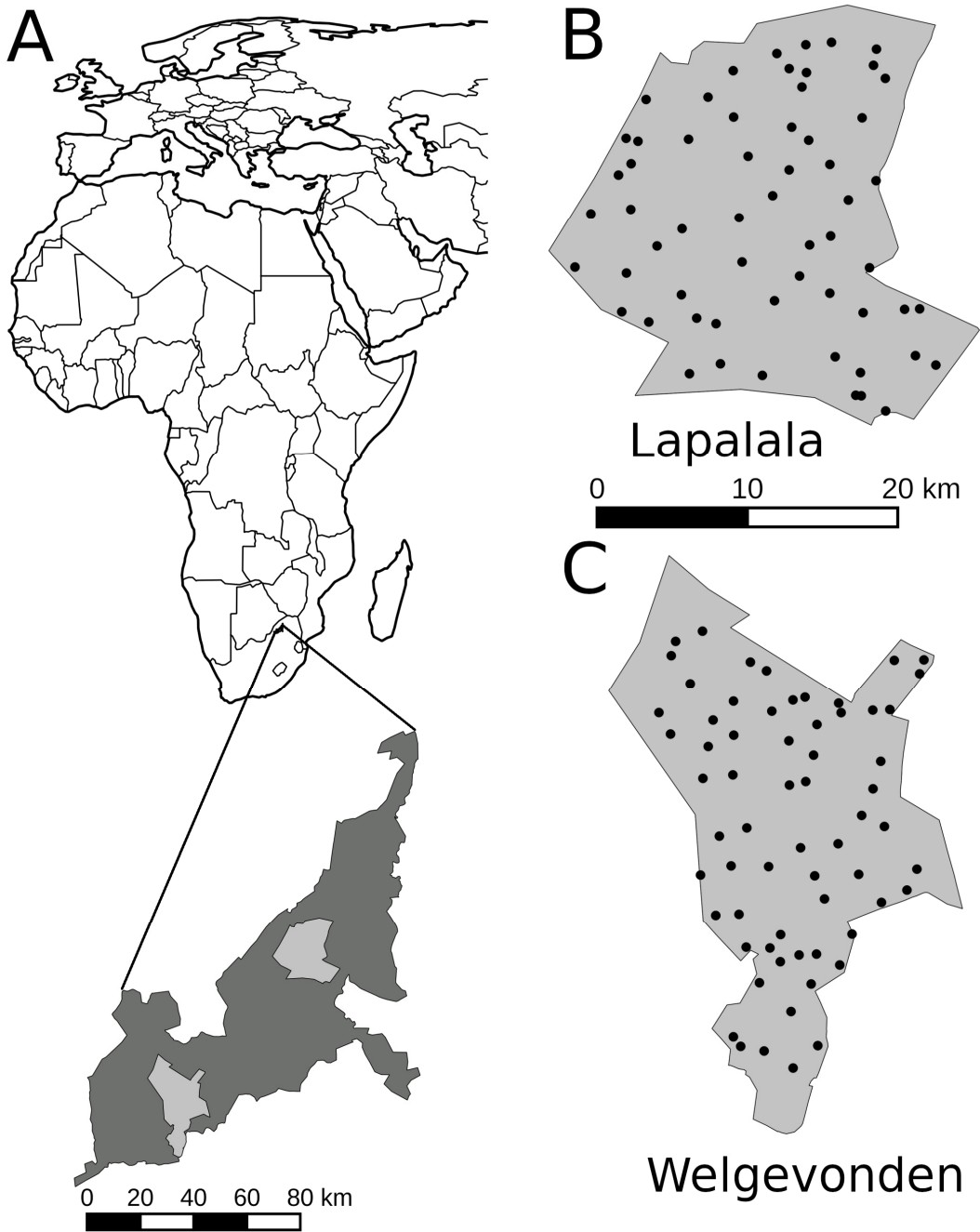

**Figure 1.** Locations of the two study areas in the Waterberg Biosphere, northern South Africa (**A**), as well as outlines of Lapalala Wilderness (**B**) and Welgevonden Game Reserve (**C**), including locations of the camera stations used for assessing relative abundance and habitat use of brown hyaenas and leopards.

The two reserves are situated less than 50 km apart and have similar sizes (Lapalala 360 km$^2$; Welgevonden 375 km$^2$), topography, vegetation and fauna [30–32]. Both reserves are privately owned.

During the time of the study Lapalala had been closed to the public for several years, whereas Welgevonden was open for tourism in the form of private lodging and game drive opportunities. The mammal fauna was similar between the reserves, both in terms of overall mammal species composition and abundance of potential prey species (Table A1), with the exception that Lapalala at the time of the study did not host lions or elephants (*Loxodonta africana*) [30]. Both reserves hosted large herbivore communities dominated by plains zebra (*Equus quagga*), blue wildebeest (*Connochaetes taurinus*), impala (*Aepyceros melampus*), greater kudu (*Tragelaphus strepsiceros*) and common warthog (*Phacochoerus africanus*). Healthy populations of black (*Diceros bicornis*) and white rhino (*Ceratotherium simum*), giraffe (*Giraffa camelopardalis*), red hartebeest (*Alcelaphus buselaphus*) and eland (*Taurotragus oryx*) also occurred. Carnivore communities consisted of leopard, brown hyaena, black-backed jackal (*Canis mesomelas*), caracal (*Caracal  caracal*), African civet (*Civettictis civetta*), African wildcat (*Felis lybica lybica*), slender mongoose (*Galerella sanguinea*), small-spotted genet (*Genetta genetta*), rusty-spotted genet (*Genetta maculata*) and honey badger (*Mellivora capensis*). During our study we only observed white-tailed mongoose (*Ichneumia albicauda*) and servals (*Leptailurus serval*) in Welgevonden. African wild dogs (*Lyacon pictus*) were occasionally present in Lapalala [33], individual cheetahs were occasionally present in both Welgevonden and Lapalala, and small groups of spotted hyaenas (*Crocuta crocuta*) were occasionally present in Welgevonden. However, observed cheetahs (*Acinonyx jubatus*) and spotted hyaenas were likely transient individuals, e.g., sub-adult males, since neither species occurred in stable populations within either of the reserves during the study period. Lions likely went extinct from the area in the early 20th century [34] but were re-introduced into Welgevonden in 1998. At the time of the study, Welgevonden hosted 2 prides and 2 vagrant males, the prides consisting of 2 adult males and 2 adult females, each, and one also of 1 sub-adult male. The prides had 2 and 4 cubs, i.e., individuals being less than 18 months old, during the study.

*2.2. Camera Surveys*

We used data from camera trapping surveys to estimate relative abundance and habitat use of brown hyaenas and leopards. The surveys were carried out from 15 May to 24 July 2009 in Lapalala and from 13 May to 12 August 2009 in Welgevonden. We placed pairs of cameras at 59 stations in Lapalala (Figure 1B) and at 58 stations in Welgevonden (Figure 1C). We placed camera stations according to a grid with 6.25 km$^2$ cell size, placing one station in each cell which gave a density of 17–20 camera traps per 100 km$^2$. The surveys were carried out with a block-wise system where stations in 13–15 cells were simultaneously surveyed for 18 to 20 days, after which the cameras were moved to a new set of 13–15 cells until we had covered the whole area [35]. This resulted in an effective surveyed effort of 71 days for Lapalala and 95 days for Welgevonden. This is short enough to satisfy population closure [36], while generating enough data for robust occupancy parameter estimates [37]. We placed the camera trap stations on vehicle roads or on animal paths to maximise the likelihood of observations. The cameras were placed 50 cm above the ground. Following Edwards [38], each camera station was classed as being placed within one of four broad habitat classes: open and closed scrubland and open and closed woodland. Two camera stations in Lapalala fell in other habitat classes and were excluded from habitat analyses.

We used a Moultrie I40 Digital Game Camera (Moultrie Feeders, Birmingham, AL, USA) except for two stations in Lapalala where we used film cameras (DeerCam DC100, Non Typical Inc., Park Falls, WI, USA; Stealth Cam MC2-GV, Stealth Cam, Grand Prairie, TX, USA; Trailmaster TM 1550, Goodson Associates Inc., Lenexa, KS, USA). The trigger mechanism was activated by movement sensors for the digital cameras and by active infrared detector beams for the film cameras. Delays between consecutive photos were set to 1 min for digital cameras and to 8 min for film cameras. Due to a slow digital camera trigger speed, we baited each camera station with a mix of rotten eggs and fermented fish to increase the chance of capturing useful pictures [35]. Such baiting has been shown not to bias the relative detection of different species [39]. Film cameras were loaded with Fujifilm ISO 400 and pictures from digital cameras were stored on SD memory cards. We visited all active stations every 4–5 days to

ensure that the cameras were still active, to check that the SD cards were not full, to change films and to replace the bait.

### 2.3. Estimation of Prey Abundance

We obtained indices of large ungulate prey abundance from total aerial counts conducted from a helicopter at each reserve. These counts are routinely carried out in both Lapalala and Welgevonden for management purposes. We only used game count data from 2008 for Lapalala since this reserve did not conduct a count during 2009, whereas we used data from surveys in both 2008 and 2009 from Welgevonden. All aerial surveys were conducted during 3 (Lapalala) and 4 (Welgevonden) days in the dry winter period (September). Surveys consisted of flying parallel transects with a helicopter using a strip width of 300 m in Lapalala and 400 m in Welgevonden. While total aerial counts may be subject to underestimation of certain species [40], it is widely regarded as a robust method for large mammals in sufficiently open terrain [41–43].

### 2.4. Estimation of Brown Hyaena and Leopard Diet

We estimated the diet of brown hyaenas and leopards by analying the content of collected faecal samples. Brown hyaena faeces were collected opportunistically along roadsides by identifying latrine and defecation sites from a motor vehicle [44], and leopard faeces were collected by driving the roads deliberately looking for samples. All samples were collected by field personnel experienced in species identification of carnivore scats. In both Lapalala and Welgevonden, faeces were collected during five periods: June to August and November 2008, August and October 2009 and March 2010. Collected faeces were placed into paper bags labelled with the reserve, sample number, date, and GPS location. The samples were left in the sun until they were completely dry. During June to August in Lapalala, all faeces at one latrine or defecation site were put in the same bag, whereas each estimated faecal unit was placed in individual bags during consecutive collection periods. However, all bags from a specific latrine site were labelled with a unique name of this location.

Each individual faeces from leopards and the content of each collected bag of brown hyaena faeces (i.e., either individual faeces or groups collected at individual latrines) were washed in small cloth bags with a 124 μm mesh in an electric washing machine [45]. The washed remains were oven dried at 70 °C for 24 h and analysed for the presence of macro remains (bone, teeth, seeds, plastic fragments) of the main diet categories as defined below. In addition, between 15 and 20 hairs of different length, texture and colour were taken from each collected sample for hair analysis. Cuticular scale imprints were made in gelatine and cross sections were made by encasing the hair in paraffin wax and then making cross sections. Each hair was identified down to the lowest taxonomical rank possible based on cross-referencing to hairs from known specimens and via the use of reference keys [46–49].

We pooled macroscopically quantified remains in the faeces into 5 broad groups: large ungulates (>200 kg), medium sized ungulates (50–200 kg), small ungulates (<50 kg), and a non-ungulate category containing chacma baboon (*Papio ursinus*), vervet monkey (*Chlorocebus pygerythrus*), Cape porcupine (*Hystrix africaeaustralis*), tortoises, rodents, carnivores and insects. We removed plant remains, non-digestible matter (plastic, charcoal, furniture pieces, rocks) and unknowns (all faeces containing unidentifiable hair and/or bone) for analyses. The size class of identified ungulate species are listed in Table A2.

### 2.5. Data Analyses

We used occupancy models based on the camera trapping data to assess the relative abundance of brown hyaenas and leopards, as well as their relative use of different habitats, while accounting for imperfect detection [50]. We used the initial formulation of occupancy proposed by MacKenzie et al. [37], which defines occupancy as the proportion of sites or area occupied by a species, given that the probability of detection may be less than one. This interpretation of occupancy has previously been suggested as a viable proxy for animal abundance, including estimations of habitat use [51].

The probability of detection is estimated from repeated observations from some fixed observation points, in our case camera stations. Both occupancy and detection can be estimated using linear covariates on a logit scale.

We fitted separate models for each species but included data from both reserves. For each model, we used reserve, habitat and their two-way interaction as covariates for the occupancy state. We evaluated the effects of these occupancy covariates using sequential likelihood ratio tests [52]. We similarly added reserve, habitat and their two-way interaction as detection covariates. However, in contrast to the covariates for the occupancy state, we used Akaike's Information Criterion (AIC) to evaluate the optimal structure for detection covariates [53]. We treated models within 2 delta AIC units as having equal empirical support [54], and we selected the least complex model if more than one model was regarded to have equal support. To improve accuracy of estimated habitat specific occupancy, we also fitted separate subset models for each reserve using only habitat class as occupancy covariate and the habitat as a detection covariate if supported by AIC values (Table A3). For both occupancy and detection, reserve and habitat were added as non-ordered factors containing 2 and 4 levels, respectively.

We used a generalised linear model with a log link and a Poisson error distribution to compare the diet composition of brown hyaenas and leopards between Lapalala and Welgevonden. We ran separate models for each species. In the models, we used the broad diet categories, reserve and their two-way interaction as predictors, and the raw frequencies of occurrences as response variable.

We used Jacob's index to estimate prey selection of ungulate species by brown hyaenas and leopard [55]. We calculated separate indices for brown hyaenas and leopards in each reserve. The index $D_{ij}$: for ungulate species i in reserve j was calculated as:

$$D_{ij} = (r_{ij} - p_{ij})/(r_{ij} + p_{ij} - 2r_{ij}p_{ij}) \tag{1}$$

where $r_{ij}$ is the proportion of prey i in the diet of brown hyaenas and leopards, respectively, in reserve j, and $p_{ij}$ is the proportion of occurrences of species i in the game count of reserve j. The index takes values from −1, indicating strong avoidance, to 1, indicating strong preference, where values close to 0 indicate a dietary use in proportion to availability. We used paired Wilcoxon signed-rank test to evaluate differences between Lapalala and Welgevonden in terms of the index values. We did separate test for brown hyaenas and leopards.

All statistical analyses were conducted using the software R version 3.6.3 compiled for the Linux environment [56] and the user defined package unmarked [57]. All mean values are given ±SE.

## 3. Results

### 3.1. Relative Abundance and Habitat Use

Predicted occupancy did not differ between populations living without and with lions for either brown hyaenas ($\chi^2 = 1.28$, df = 1, $p = 0.257$) or leopards ($\chi^2 = 0.37$, df = 1, $p = 0.544$), although predicted occupancy was lower for the populations living without than with lions for both brown hyaenas (without lions $0.66 \pm 0.08$; with lions $0.79 \pm 0.07$) and leopards (without lions $0.64 \pm 0.16$; with lions $0.74 \pm 0.15$) (Figure 2). We made a total of 73 temporally independent (i.e., not with the same 24-h period) observations of brown hyaenas in the population living without and 96 in the population living with lions, and made 21 and 39 observations of leopards in the populations living without and with lions, respectively. These observations were made at 33 and 39 separate camera station for brown hyaenas and at 18 and 24 stations for leopards. Average detection probabilities for brown hyaenas living without and with lions were 0.11 ($\pm$ 0.01) and 0.09 ($\pm$ 0.01), and average detection probabilities for leopards were 0.03 ($\pm$ 0.01) and 0.04 ($\pm$ 0.01).

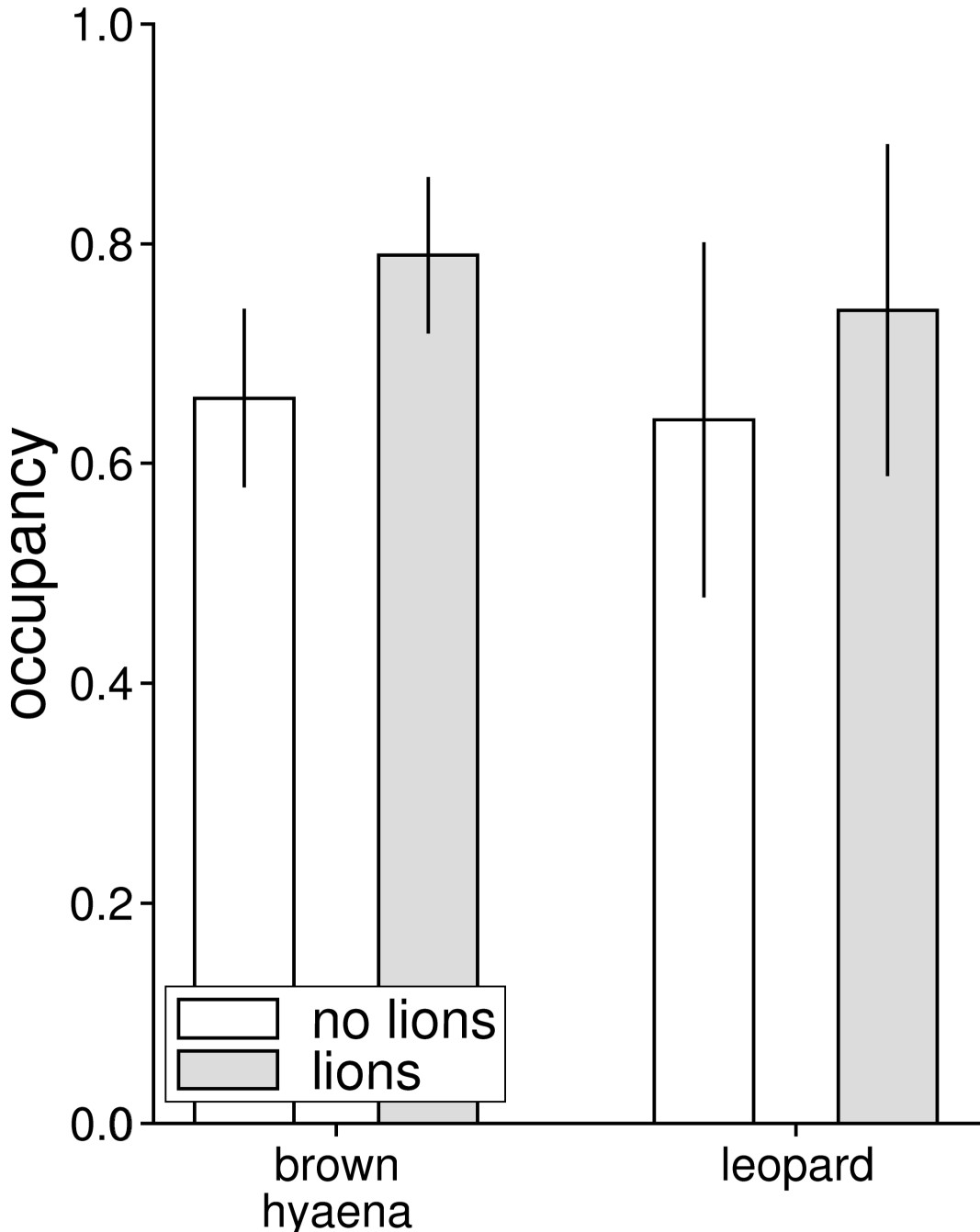

**Figure 2.** Predicted probability of occupancy for brown hyaenas and leopards in a reserve without (Lapalala) and with (Welgevonden) lions present. Occupancy estimates are based on single species occupancy models, which accounts for imperfect detection, based on data from camera trap surveys.

The estimated occupancy of different habitat classes did not differ between populations living without and with lions for either brown hyaenas ($\chi^2$ = 0.98, df = 3, $p$ = 0.807) or leopards ($\chi^2$ = 0.09, df = 3, $p$ = 0.993). However, brown hyaenas living without lions had higher estimated occupancy

in closed scrubland and open woodlands than in open scrubland and closed woodland, whereas there were less pronounced habitat differences for brown hyaenas living with lions (Figure 3A). Leopards living without lions had higher estimated occupancy in scrublands than in woodlands, whereas leopards living with lions had higher estimated occupancy in open scrubland and open woodland than in closed shrubland and woodland (Figure 3B).

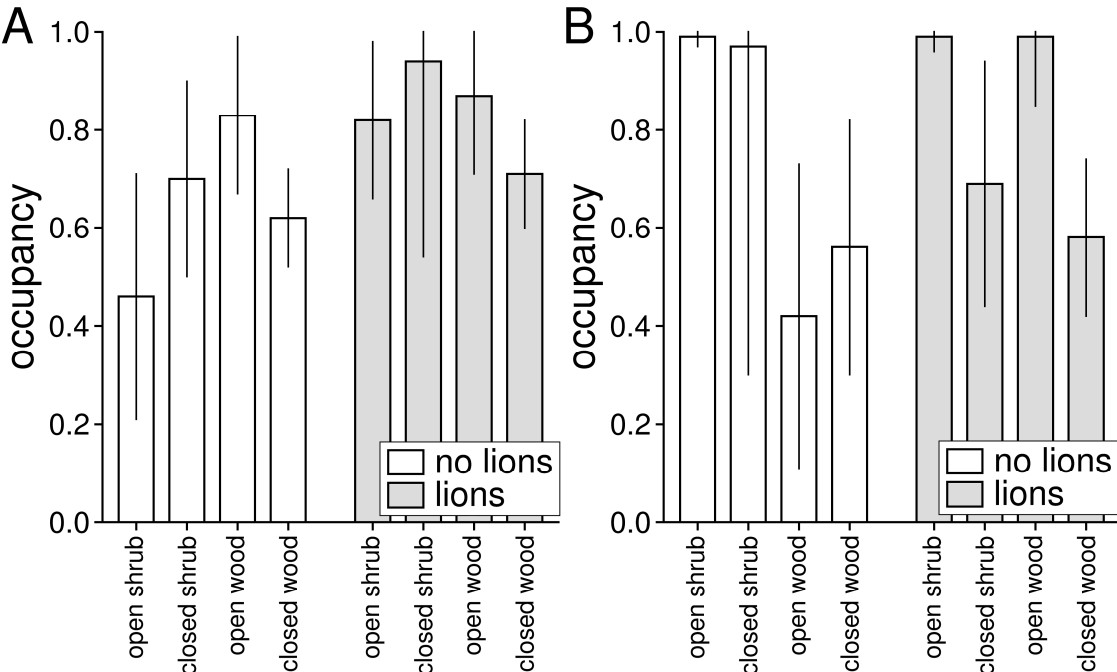

**Figure 3.** Estimated as probability of occupancy of brown hyaenas (**A**) and leopards (**B**) in four broad habitat classes in a reserve without (Lapalala) and with (Welgevonden) lions. Occupancy estimates were estimated from single species occupancy models based on camera trapping data.

The optimal occupancy model using brown hyaena data from both Lapalala and Welgevonden included reserve and habitat as detection covariates, but not their interaction (Table A3). The optimal model had a delta AIC value 1.92 units above the model with the lowest AIC value but contained three parameters less. The optimal subset model only using brown hyaena data from Lapalala included habitat as a detection covariate, whereas the optimal subset model using brown hyaena data only from Welgevonden did not include detection covariates (Table A3). The optimal model using leopard data from both Lapalala and Welgevonden did not include detection covariates, as this model was within two AIC units from the model with the lowest AIC value but had four less parameters (Table A3). Similarly, the optimal subset model using leopard data only from Lapalala did not include detection covariates, whereas the optimal subset model using leopard data only from Welgevonden did include habitat as a detection covariate.

### 3.2. Diet and Prey Selection

Broad diet composition differed between brown hyaenas living without and with lions ($\chi^2 = 23.15$, df = 3, $p < 0.001$), and there was a trend for the diet composition also to be different for leopards living without and with lions ($\chi^2 = 7.00$, df = 3, $p = 0.072$). Brown hyaenas living without lions fed to a larger extent on medium-sized and small ungulates compared to hyaenas living with lions, and to a lesser extent on large ungulates and non-ungulate prey (Figure 4A). Leopards living without lions appeared to

have fed more on large ungulates and medium sized ungulates compared to leopards living with lions, and less on non-ungulate prey (Figure 4B).

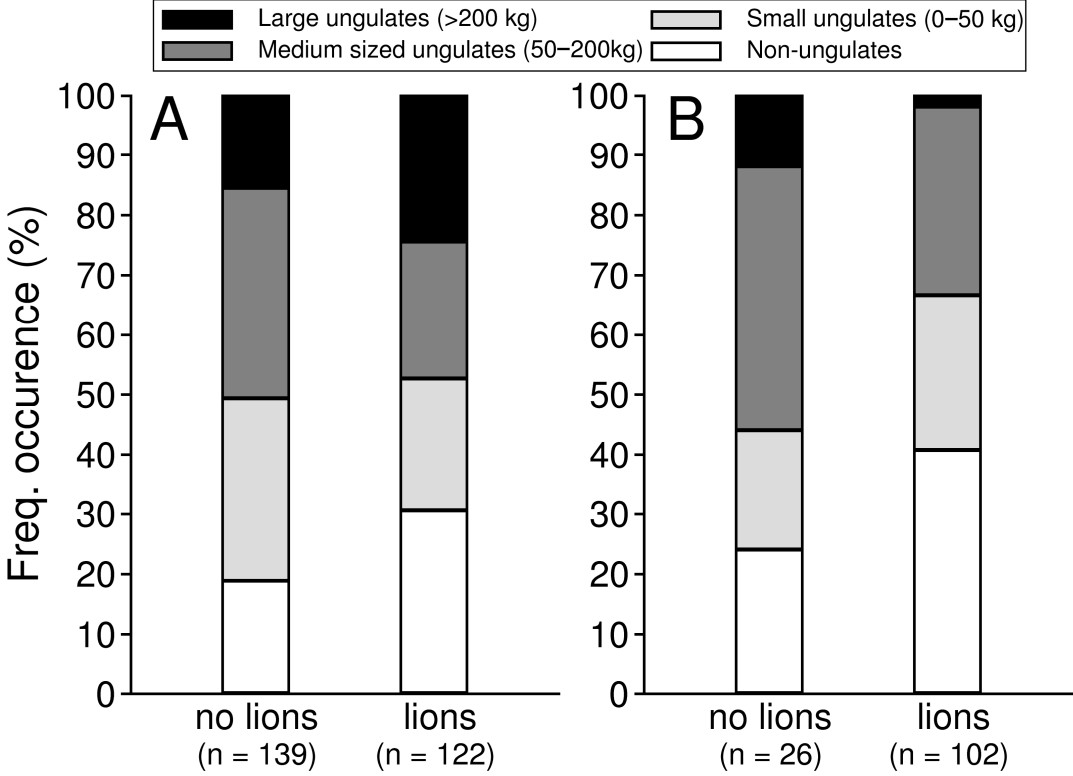

**Figure 4.** Diet composition of brown hyaenas (**A**) and leopards (**B**) in a reserve without (Lapalala) and with lions (Welgevonden).

Prey selection of ungulates did not differ between populations living without and with lions for either brown hyaenas ($V = 37$, $n = 10$, $p = 0.375$) or leopards ($V = 32$, $n = 10$, $p = 0.286$). Brown hyaenas fed less on large ungulates than expected by abundance, and instead showed a preference for medium sized and small ungulates (Figure 5A). Leopards similarly fed less on large ungulates than expected by abundance, whereas they showed more variable preferences for medium sized and small ungulates (Figure 5B). In particular, leopards showed a preference for bushbuck and common duiker, and in Welgevonden also for klipspringer and nyala.

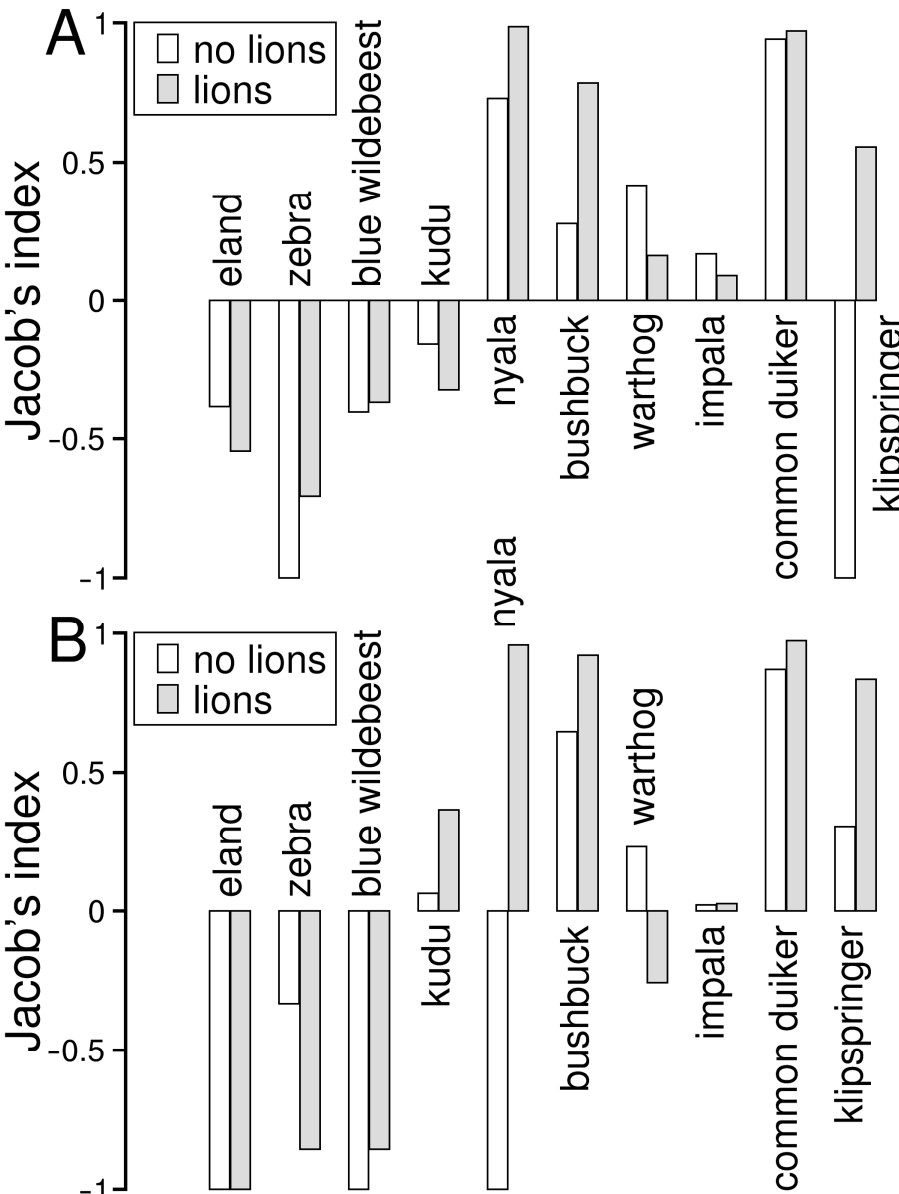

**Figure 5.** Selection of main prey species, quantified as Jacob's index, of brown hyaenas (**A**) and leopards (**B**) in a reserve without (Lapalala) and with lions (Welgevonden). Jacob's index ranges from −1, which indicates strong avoidance, to 1, indicating strong preference.

## 4. Discussions

Our study did not find strong support for either facilitative or competitive effects of lions on two contrasting large carnivores, the brown hyaena and the leopard. Although our data suggested differences in broad diet compositions for brown hyaenas and leopards living with and without lions, we did not observe different preferences of species within their main prey group, ungulates. In addition, we found limited evidence for differences in abundance or habitat use. These results agree with other recent findings suggesting that lions may not necessarily impose the large effects of sympatric predators [21,23,32,58], although such effects have been observed to occur [24]. Hence, despite strong claims for the ecological significance of apex predators [59], we argue that our study adds to a mounting body of literature suggesting that sweeping, landscape level effects of apex predators, particularly those extending beyond consumptive effects on prey populations, may not be as strong or common as previously thought [31,32,60–62]. Instead, predator effects on sympatric species appear to be highly context dependent, for both sympatric predators and prey [63,64].

Although we predicted that a scavenger like the brown hyaena would have higher abundance in the presence of lions due to increased carcass availability, and concurrently that competition would suppress leopard abundance in the presence of lions, we did not observe such effects. In addition, we did not find any strong evidence for shifts in the use of broad habitat classes for leopards. We suggest that these results imply that facilitative or competitive interactions with lions were either absent or not strong enough to permeate into differences in population sizes or the distribution of brown hyaenas and leopards across the landscape. We have previously observed a lack of competition effects on the diel activity of sympatric carnivores in this same system [63], and our results also agree with studies in other areas of southern Africa [21,23,58,65]. Instead of competition, we suggest that these results indicate that resource availability and distribution may have been the strongest drivers of brown hyaena and leopard abundance and distribution [23,65]. Such an interpretation implies that these large carnivores, at least to some extent, are regulated by bottom up forces, and put further emphasis on the need to evaluate under what circumstances ecological communities are regulated by primary productivity and resource supply versus biotic interactions among community members [66].

For both brown hyaenas and leopards, our results pointed to differences in overall diet composition between populations living without and with lions. However, despite ungulates forming the main diet categories for both of these predator species, we did not observe any differences in the prey preferences for ungulate species between populations living without and with lions. Facilitative interactions within carnivore communities have been recognised as important mechanisms structuring ecosystems [67,68]. Our result lends support to previous studies suggesting that brown hyaenas utilise carrion provided by lions [25,69], but since we detected no differences in brown hyaena abundance it is unclear what ecological consequences such carcass provision may have had. Our results further suggest that leopards in the presence of lions used non-ungulate prey instead of large ungulates, in agreement with predictions based on competitive interactions. However, as with potential facilitative interactions between lions and brown hyaenas, it is unclear if such competitive interactions resulted in any broader effects on leopard ecology, since we did not detect differences in either abundance, habitat use or ungulate prey selection. Weak dietary competition has previously been suggested between leopards and lions [22], and we highlight that identifications of the conditions under which intra-guild processes influence carnivore communities should be a prioritised component of further carnivore community studies.

While we regard our results as robust, we recognise some limitations of our study. First, on a landscape scale it has an effective sample size of one [70]. Therefore, broader generalisations beyond our study system may not be appropriate. However, primary studies based on direct observations, such as this, need to form the backbone of our inquiries into the reality we live in, even if they have limited statistical sample sizes compared to data accumulated over time or space [71]. We also appreciate that limited observations could have caused our negative results, either in terms of camera captures or collected scat samples. However, our occupancy analyses of leopards, for which we had fewer observations than

brown hyaenas, agree with previous mark–recapture models in this area [35]. Furthermore, although we estimated low detection probabilities, our sampling duration exceeds durations estimated as generating acceptable occupancy estimates in simulations using comparable levels of detection [37]. Supporting this observation is the relatively small standard errors around our occupancy estimates for each reserve and species, suggesting that our quantified differences between the reserves were robust. We similarly argue that our comparisons of diet between the reserves are robust for both species. Although improving sample sizes in scat analyses increase the precision of dietary estimates and increase the likelihood of identifying unusual prey items, even limited sample sizes accurately identify the proportion of common prey classes [72]. We do, however, call for caution in terms of interpreting our prey selection indices quantitatively between prey species. We based these indices on aerial counts, which likely underestimate small species relative to large ones, and are also likely to bias counts towards species favouring open habitats over woodlands. However, ungulate prey communities were counted with similar methodologies on both reserves. Therefore, we regard our contrasts in prey selection between reserves as robust since any bias should be consistent across reserves.

While we did observe differences in broad diet compositions in agreement with facilitative effects of lions on brown hyaenas and competitive effects on leopards, our results suggest that such potential facilitative and competitive interactions were not strong enough to permeate into differences in abundance, habitat use, and prey selection for populations of brown hyaenas living without and with lions. We therefore interpret our results as further support for limited or context dependent ecological consequences of apex predators, at least on broad landscape scales, and call for further studies identifying how and under which conditions lions and other apex predators influence the ecosystems they live in.

**Author Contributions:** Conceptualisation, F.D.; Data curation, L.S. and F.D.; Formal analysis, J.B. and F.D.; Funding acquisition, F.D.; Investigation, J.B., L.S. and F.D.; Methodology, J.B., L.S. and F.D.; Project administration, L.S. and F.D.; Resources, L.S. and F.D.; Supervision, M.S. and F.D.; Visualisation, F.D.; Writing—original draft, J.B. and F.D.; Writing—review and editing, M.S. and L.S. All authors have read and agreed to the published version of the manuscript.

**Funding:** This research was funded by the National Geographic/Wait's Foundation (grant number W32-08 to FD), the National Research Foundation in South Africa (grant numbers SFP2008072900003 to FD, IFR2011032400087 and UID 115040 to LS), and the Ministry of Economy and Competitiveness in Spain (grant number RYC2013-14662 to FD).

**Acknowledgments:** We are grateful to managers and staff at Lapalala Wilderness and Welgevonden Game Reserve for permission to carry out the research and for logistical support. Lydia Belton kindly assisted with washing and processing of faecal samples.

**Conflicts of Interest:** The authors declare no conflict of interest.

## Appendix A

**Table A1.** Number of animals observed during aerial game counts in Lapalala and Welgevonden. The game counts were carried out from helicopters in September 2008 and 2009 (only Welgevonden).

| Common Name | Latin Name | Lapalala | Welgevonden | |
|---|---|---|---|---|
| | | 2008 | 2008 | 2009 |
| Impala | *Aepyceros melampus* | 1090 | 695 | 701 |
| Burchell's zebra | *Equus burchellii* | 1156 | 472 | 573 |
| Blue wildebeest | *Connochaetes taurinus* | 409 | 469 | 557 |
| Greater kudu | *Tragelaphus strepsiceros* | 606 | 183 | 166 |
| Common warthog | *Phacochoerus africanus* | 293 | 220 | 144 |
| Eland | *Taurotragus oryx* | 200 | 158 | 156 |
| Waterbuck | *Kobus ellipsiprymnus* | 206 | 140 | 123 |
| Klipspringer | *Oreotragus oreotragus* | 123 | 44 | 34 |
| Giraffe | *Giraffa camelopardalis* | 67 | 30 | 24 |
| Bushbuck | *Tragelaphus scriptus* | 106 | 6 | 4 |
| Mountain reedbuck | *Redunca fulvorufula* | 44 | 28 | 20 |
| Common duiker | *Sylvicapra grimmia* | 35 | 6 | 2 |
| Nyala | *Tragelaphus angasii* | 30 | 1 | 4 |

**Table A2.** Size classes of ungulates identified in brown hyaena and leopard diets that was used for broad diet assessments.

| Common Name | Latin Name | Size Class |
|---|---|---|
| Bushbuck | *Tragelaphus scriptus* | Small (<50kg) |
| Common duiker | *Sylvicapra grimmia* | Small (<50kg) |
| Klipspringer | *Oreotragus oreotragus* | Small (<50kg) |
| Greater kudu | *Tragelaphus strepsiceros* | Medium (50–200kg) |
| Impala | *Aepyceros melampus* | Medium (50–200kg) |
| Nyala | *Tragelaphus angasii* | Medium (50–200kg) |
| Common warthog | *Phacochoerus africanus* | Medium (50–200kg) |
| Blue wildebeest | *Connochaetes taurinus* | Large (>200kg) |
| Burchell's zebra | *Equus burchellii* | Large (>200kg) |
| Eland | *Taurotragus oryx* | Large (>200kg) |

**Table A3.** Included data, occupancy and detection covariate structures, number of parameters as well as delta ($\Delta$) Akaike's Information Criterion (AIC) values for several candidate occupancy models of brown hyaenas and leopards in Lapalala and Welgevonden. Models within 2 $\Delta$ AIC units were regarded as having equal support. The candidate models are ordered by relative empirical support (indicated by $\Delta$ AIC values), and, if more than one candidate had approximately equal empirical support (i.e., had 2 AIC units or less compared to the model with the lowest AIC), also by the lowest number of parameters. If several models had equal support, the one with the lowest number of parameters was used for final data interpretation. The covariates are non-ordered factors with 2 (Reserve) and 4 (Habitat) levels.

| Species | Data | Occupancy Covariates | Detection Covariates | Par. | $\Delta$ AIC |
|---|---|---|---|---|---|
| Brown hyaena | Both | Reserve x Habitat | Reserve + Habitat | 13 | 0 |
| Brown hyaena | Both | Reserve x Habitat | Reserve x Habitat | 16 | 1.92 |
| Brown hyaena | Both | Reserve x Habitat | None | 9 | 25.35 |
| Brown hyaena | Both | Reserve x Habitat | Reserve | 10 | 27.35 |
| Brown hyaena | Lapalala | Habitat | None | 5 | 0.92 |
| Brown hyaena | Lapalala | Habitat | Habitat | 8 | 0 |
| Brown hyaena | Welgevonden | Habitat | Habitat | 8 | 0 |
| Brown hyaena | Welgevonden | Habitat | None | 5 | 6.40 |
| Leopard | Both | Reserve x Habitat | None | 9 | 2.00 |
| Leopard | Both | Reserve x Habitat | Reserve | 10 | 1.26 |
| Leopard | Both | Reserve x Habitat | Reserve + Habitat | 13 | 0 |
| Leopard | Both | Reserve x Habitat | Reserve x Habitat | 16 | 3.57 |
| Leopard | Lapalala | Habitat | None | 5 | 0 |
| Leopard | Lapalala | Habitat | Habitat | 8 | 5.07 |
| Leopard | Welgevonden | Habitat | Habitat | 8 | 0 |
| Leopard | Welgevonden | Habitat | None | 5 | 7.91 |

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
