# Peer review of "Facilitation or Competition? Effects of Lions on Brown Hyaenas and Leopards"

_diversity, doi:10.3390/d12090325_

Round 1
Reviewer 1 Report
Please see attached file for detailed comments

Author Response
Please see the attached pdf file for our detailed responses

Reviewer 2 Report
Line 62 – should be “small- and medium-sized prey”
L63 – “medium-sized”
L71 – “lion” should be “lions”
L77 – “competitions” should be “competition”
L78 - “leopard” should be “leopards”
L79 – “smaller sized” should be “smaller-sized”
L90 – change areas to km2, not hectares (scale to study animals)
L92 – “but Lapalala had during the time of study been closed” should be “but during the time of the study, Lapalala had been closed”
L105-109 – “African wild dogs (Lyacon pictus) were occasionally present in Lapalala [33], individual cheetahs were occasionally present in both Welgevonden and Lapalala, and small groups of spotted hyenas (Crocuta crocuta) were occasionally present in Welgevonden. However, observed cheetahs and spotted hyaenas were likely transient individuals, e.g., sub-adult males, since neither species occurred in stable populations within either of the reserves [34].” Why not just included from “33. Ramnanan, R., Swanepoel, L., Somers, M. The diet and presence of African wild dogs (Lycaon pictus) on private land in the Waterberg region, South Africa”:
Welgevonden Lapalala
(2008–2010) (2008–2010)
Brown hyaena 20.83 12.63
Leopard 3.54 3.00
Lion 2.30 0.00
Spotted hyaena 1.25 0.09
Cheetah 0.17 0.00
Wild dog 0.00 0.73
although Ramnanan et al. does not talk about spotted hyena presence in “small groups”. What other evidence is there in the upublished and thus not available: “34. Greco, I., Chizzola, M., Meloro, C., Swanepoel, L., Tamagnini, D., Dalerum, F. Similarities in size, morphology and diel activity between lions and sympatric carnivores. J. Zool.under review.” How much does this manuscript support or overlap what you are trying to say in this manuscript?
L140 – How do these data compare with your 3604 images of prey animals from camera trapping results in [33]? Why not also use these data to see if you get the same results?
L197-198 – why no detection covariate for leopards in Lapalala?
L220 – “24-hour”
L233 – Turn Figure 4 into a Table with all data and sample sizes.
L292-308 – I like these caveats – thank you.
